# Evaluation of Radiation Exposure Due to Dental Radiographs Taken during Endodontic Treatment Sessions in Young Permanent Teeth

**DOI:** 10.3390/medicina58121822

**Published:** 2022-12-11

**Authors:** Ebru Akleyin, Yasemin Yavuz

**Affiliations:** 1Department of Peadiatric Dentistry, Faculty of Dentistry, Dicle University, Diyarbakır 21010, Turkey; 2Department of Restorative Dentistry, Faculty of Dentistry, Harran University, Urfa 63000, Turkey

**Keywords:** young permanent teeth, endodontic treatment, dental radiograpy

## Abstract

*Background and Objectives*: The aim of this study was to retrospectively analyze the duration of treatment and the number of dental radiographs taken during endodontic treatment (endo-t) of young permanent teeth (YPT). *Materials and Methods*: Age, gender, affected tooth number, apex status, duration of treatment and dental radiographs taken during this period were retrospectively evaluated in pediatric patients aged 6–15 years who presented to the pedodontic department for endo-t of anterior YPT. Data were analyzed with Kruskal Wallis H, Dunn and Pearson chi-square tests. *p* < 0.05 was accepted as statistically significant in all analyses. *Results*: Anterior endo-t was initiated in 471 of 9,200 pedodontic patients between the ages of 6 and 15 years who applied to our university. The reason for seeking treatment was caries (8.1%) and trauma (91.9%). It was observed that 59% of the teeth had an open apex and 45.7% had a closed apex. A total of 1893 periapical radiographs (Per-R) and 245 panoramic radiographs (Pan-R) were taken from 471 pediatric patients during the treatment period. Total number of dental radiographs was 2138 with 4.5 per patient. Number of Per-R was lower in patients whose treatment was completed in a single session (*p* < 0.001). There was no difference in the number of Pan-R with respect to duration of treatment (*p* = 0.560). *Conclusions:* In children, the number of Per-R significantly increased with prolonged duration of treatment encompassing multiple sessions for endo-t procedures of anterior YPT, decided based on the apex and lesion status of the affected tooth. Especially in long-term traditional apexification treatments, treatment should be carried out in children with the ALADAIP (As Low As Diagnostically Achievable being Indication-oriented and Patient-specific) principle in mind.

## 1. Introduction

Radiographic examination is frequently used in dental practice as one of the main diagnostic methods [1]. As is well known, ionizing radiation is a mutagen and carcinogen for cells. Exposure to high doses of radiation can pose a greater risk of DNA damage in children, especially since young and rapidly growing immature tissues are more radiosensitive. Although the risk is low in low-level diagnostic exposures, it is still greater than zero; therefore, there is no absolute safety limit for the dose [2]. Furthermore, radiation risk is directly proportional to the frequency of exposure to X-rays [3]. Even if the radiation dose is low in intraoral and panoramic radiographs taken during dental radiographic examination, repeated examinations cause unnecessary radiation exposure [4]. To protect dental staff and patients from unnecessary radiation exposure and potential damage caused by ionizing radiation, various guidelines have been developed. These guidelines advise on the basic principles of radiation protection: “justification, limitation and optimization”. However, these guidelines may or may not be applied during clinical practice [5]. According to the most recent EAPD guidelines, it is essential that dental radiographic examination in the young population follows the radiologic principles of individual patient-specific justification. Detailed clinical examination, the patient’s ability to cooperate, information from previous radiographs, and the use of alternative non-radiographic examination options should be the main factors for deciding on the use of radiography. In children, use of radiography should be optimized according to the ALADAIP principle, which aims to limit radiation exposure [6].

In the treatment of immature permanent teeth with necrotic pulp, traditional endo-t’s are difficult for clinicians to apply due to the open apex, wide root canal, and fragile root walls. The treatment for these teeth may involve multiple visits, depending on the apex opening of the teeth, the method of apexification, the material used, the presence of periapical pathology, the frequency of replacement of the material used, and the age of the patient [7].

Two-dimensional dental radiographs are the most commonly used methods for differential diagnosis in the evaluation of root development status and periapical pathologies of young permanent teeth (YPT). In light of the above, the aim of this study was to evaluate the duration of endo-t in YPT, and the number of dental radiographs taken during sessions, and to present a new perspective to radiographic studies.

## 2. Materials and Methods

In this retrospective study, the clinical records of 9200 patients aged 6–15 who were admitted to Dicle Faculty of Dentistry, Department of Pedodontics. All upper and lower incisors with complete endo-t were included in the study. During the treatment period, initial, interim and final radiographs of all teeth receiving endo-t were obtained digitally as Per-R or Pan-R.

For each patient, age, gender, position of the incisors with endo-t, cause of endo-t (caries, trauma), apex status before treatment (open, closed), duration of endo-t (single session, 1–6 months, 6–12 months, over 12 months), and number of dental radiographs taken during this period (Per-R, Pan-R) were recorded on a customized form. Apexification treatment methods applied in teeth with open apex were determined (calcium hydroxide and MTA apexification, regenerative endo-t). In the traditional endo-t of patients treated in a single session, the canal was filled with guta-percha. The recorded dental radiographs of the patients, including all interim sessions from the first diagnostic radiograph to the final radiograph taken after canal filling, and the treatment completion times were statistically analyzed. Data were analyzed with SPSS (IBM, v23, Armonk, NY). Kolmogorov Smirnov test showed that the data were not normally distributed. The Kruskal Wallis H test was used to compare variables between the groups and multiple comparisons were made with Dunn’s test. Pearson chi-square was used to examine the relationship between categorical variables with respect to study groups and Bonferroni correction was performed. Descriptive statistics were presented as median, minimum, maximum, frequency, and percentage. *p* < 0.05 was accepted as statistically significant in all analyses. The design of this retrospective study was approved by the Dicle University Dental Ethics Committee (2022–37). 

## 3. Results

The study included 471 pediatric patients aged 6–15 years who completed endo-t in the upper and lower incisors. Of these patients, 39.5% were female and 66.5% were male. The reason for referral to endo-t was caries in 8.1% and trauma in 91.9% of the patients. In all patients who underwent endo-t, 59% of the incisors had open apex and 41% had closed apex. When all incisors were evaluated, endo-t was most frequently performed on the maxillary right upper first central tooth (73.2%). In 45% of the patients, endo-t was completed in a single session. Duration of treatment was 1–6 months in 24.2%, 6–12 months in 16.8%, and 12 months or more in 9.3% of the patients. A total of 2138 dental radiographs, 1893 (88.5%) Per-R and 245 (11.5%) Pan-R, were taken from 471 patients (Table 1).

The number of teeth with open apex with endo-t completed in a single session or between 1–6 months and the number of teeth with closed apex teeth with endo-t completed in a single session were significantly higher compared to longer duration of treatment (Table 2).

A significant difference was found in the median Per-R count with respect to duration of treatment (*p* < 0.001). The median Per-R value was 3 for a single session, 4 for 1–6 months, 4 for 6–12 months, and 4 for 12 months or more. Median Per-R for single-session was lower compared to the other groups. No significant difference was found in median Pan-R value with respect to duration of treatment (*p* = 0.560). In contrast, a significant difference was found in total number of radiographs with respect to the duration of treatment (*p* < 0.001). The median value of total radiographs taken was 3 for a single session, 4 for 1–6 months, 4 for 6–12 months, and 4.5 for 12 months or more. While there was no difference between single session and 1–6 months, number of radiographs taken in a single session was lower compared to 6–12 months and 12 months or more (Table 3).

## 4. Discussion

This study was planned to identify gaps in treatments where young people have an increased need for radiation protection. According to the results obtained, a total of 2138 radiographs were taken during clinical endo-t of YPT in 471 pediatric patients. To the best of our knowledge, there is no such study in the literature.

It was reported that an estimated 500 million intraoral whole-mouth radiographic procedures performed in the United States in 2006 was almost twice the total number of traditional medical radiographic examinations [8]. Therefore, one of the most common radiographs performed in children aged 6–15 years is likely dental radiographs taken in cases of dental trauma. Especially in head and neck cancers, the need to create an information system that collects all radiation data to evaluate radiation outcomes should be emphasized.

Up to 25% of school-age children and 33% of adults before the age of 19 are exposed to dental trauma [9]. Pulp necrosis, which is common in children, especially in anterior teeth as a result of trauma, affects the root development of the teeth, resulting in brittle root walls and an open apex. The absence of an apical stop in the roots makes traditional endo-t difficult [10,11]. In the present study, endo-t was most frequently performed in children between the ages of 6 and 15 on teeth with open apex as a result of trauma, which was consistent with the literature.

In necrotic teeth with an open apex, calcium hydroxide has been used for many years for apexification, the induction procedure of a calcified apical barrier, but it requires long treatment with multiple sessions. In recent years, apexification treatment with mineral trioxide aggregate (MTA) instead of calcium hydroxide has emerged as a successful treatment option to create an apical plug in a single session. Regenerative endodontics, on the other hand, is a treatment modality involving tissue engineering, biomimetic scaffolding, and revascularization of damaged pulp tissue with bioactive growth hormones. It is performed within four weeks after the first session and requires long follow-up [7,12]. In the present study, endo-t sessions in children taking 12 months or more were most frequently performed on teeth with open apex. However, the fact that all apexification protocols were performed as part of pediatric clinical practice where the study was conducted is strong evidence of case-specific treatment.

According to the European Commission radiation protection guidelines, although some types of electronic apex locators are reliable for apical determination, periapical radiography is still usually required for determining working length. If there is doubt about the integrity of the apical stenosis, radiography with the main gutta-percha cone before the final control is recommended. Sometimes two (or more) radiographs may be required [13]. In the present study, the number of Per-R’s was higher in teeth with open apex.

Intraoral images may be sufficient to assess the type, location and severity of dento-alveolar injuries [14]. In a survey of 255 Swedish orthodontists, it was found that the participants preferred Per-R over Pan-R for periapical lesions [15]. In a study evaluating the concordance between Pan-R and Per-R, it was reported that Pan-R alone was not sufficient to diagnose periapical lesions, marginal bone loss, and caries and intraoral radiograph was a better option when limited areas needed to be examined [16]. In the present study, Pan-R was usually performed at the initial diagnosis for root canal treatment. Özata et al. [17] found that Pen-R (80%) was more accurate in canal length measurement compared to dental tomography (60%). It has been found that cone-beam computed dental tomography (CBCT) imaging, which is increasingly used in children, is not used in trauma patients. In the present study, Per-R was mostly preferred for the endo-t of YPT caused by trauma and CBCT was not preferred at all. This may be due to the fact that traditional two-dimensional imaging methods are sufficient for diagnosis and treatment.

Hujoel et al. found that the average number of radiographs taken per patient with at least one year of orthodontic treatment was 33.7 [18]. In the present study, the average number of radiographs taken for single tooth treatment was found to be 4.5 compared to orthodontic treatments lasting 3 to 4 years, and it was thought that the recommendations in the relevant guidelines were insufficient.

It has been reported that educators perform more radiographs than private practitioners [19,20]. Since the present study was conducted in a university hospital, making such an evaluation regarding the treatments performed by educators and students was deemed unbefitting.

The ALARA (As Low As Reasonably Achievable) principle, introduced by the International Commission on Radiological Protection (ICRP) in 1977, is used to optimize radiation doses. The ALARA principle involves keeping radiation exposure to the lowest level reasonably achievable in all irradiations [21,22]. In 2014, the National Council on Radiation Protection and Measurements (NCRP) changed from ALARA to ALADA (As Low as Diagnostically Acceptable) to emphasize the importance of dose optimization in medical diagnostic imaging [23]. This change involves optimization aimed at using the lowest radiation dose consistent with adequate image quality [24]. The DIMITRA project (Dentomaxillofacial paediatric imaging: an investigation toward low-dose radiation induced risksdentomaxillofacial) is focused on optimizing pediatric doses. The DIMITRA association proposes to move from the ALARA and ALADA principles to the ALADAIP principle in order to approach the risks involved from different interrelated perspectives in a multidisciplinary manner [25]. The ALADAIP principle aims to define the appropriate balance between dose and image quality in an age- and indication-oriented manner [26]. According to the results of this study, the use of the ALADAIP principle, which is generally used in children for CIBT, can be recommended for the endo-t procedure of YPT.

Based on the results obtained in the present study, the following recommendations can be made together with radiation protection protocols in the endodontic treatment of teeth with open apex:The determination of the radiographs required, especially in trauma patients, should be made on a case-specific basis by the physician after a detailed clinical examination. The dentist should be the only authorized and responsible person in this regard.Previous radiographs must be utilized between sessions.Radiation-free auxiliary protocols such as apex locator can be applied.Treatment protocols that can reduce the number of sessions should be determined.All radiographic images obtained between sessions should be recorded.

## 5. Conclusions

Periapical radiographs taken during the treatment significantly increase as the number of endo-t sessions for YPT increase. The determination of the radiographs required in apexification treatments should be evaluated by the physician on a case-specific basis and treatment principles based on the ALADAIP principle should be followed.

## Figures and Tables

**Table 1 medicina-58-01822-t001:** Frequency and percentage values for treatment variables.

	Frequency (n)	Percentage (%)
**Gender**		
Female	186	39.5
Male	285	60.5
**Tooth number**		
11	345	73.2
12	30	6.4
13	1	0.2
21	84	17.8
22	6	1.6
31	1	0.2
32	1	0.2
41	1	0.2
42	1	0.2
**Reason for endodontic treatment**		
Trauma	433	91.9
Caries	38	8.1
**Developmental status of the apex**		
Open	278	59
Closed	193	41
**Treatments applied on teeth with open apex**		
Apexification with CaOH	134	48.2
Apexification with MTA or Biodentin	95	34.1
Regenerative endodontic treatment	49	17.6
**Number of radiographs taken**		
Panoramic radiograph	245	11.5
Periapical radiograph	1893	88.5
**Duration of endodontic treatment**		
Single session	212	45.7
1–6 Months	114	24.2
6–12 Months	79	16.8
12 months and over	44	9.3

**Table 2 medicina-58-01822-t002:** Comparison of teeth with open apex and closed apex with respect to duration of treatment.

	Duration of Treatment				Test Statistic	*p* *
Developmental Status of the Apex						
Open apex	89 (32.48) ^a^	89 (78.1) ^b^	64 (23.36) ^b^	36 (13.14) ^b^	69.5	<0.001
Closed apex	129 (75.44) ^a^	30 (17.54) ^b^	20 (11.70) ^b^	14 (8.19) ^b^		

* Pearson Chi-Square test; ^a,b^ There is no difference between treatment durations indicated with the same letter in the same row.

**Table 3 medicina-58-01822-t003:** Comparison of the number of periapical and panoramic radiographs taken during endodontic treatment sessions.

	Number of Dental Radiographs				Test Statistic	*p* *
Periapical	3 (1–11) ^a^	4 (1–13) ^b^	4 (1–18) ^b^	4 (1–12) ^b^	29.377	0.56
Panoramic	0 (0–3)	0 (0–2)	0 (0–2)	0 (0–3)	2.062	
Total	3 (2–13) ^a^	4 (2–15) ^ab^	4 (2–20) ^b^	4.5 (2–12) ^b^	21.162	

* Kruskal-Wallis H test; ^a,b^ There is no difference between treatment durations indicated with the same letter.

## Data Availability

Not applicable.

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
