# Peer review of "Evaluation of Radiation Exposure Due to Dental Radiographs Taken during Endodontic Treatment Sessions in Young Permanent Teeth"

_medicina, 2022, doi:10.3390/medicina58121822_

Round 1

Reviewer 1 Report

The paragraph in the discussion section lines 188-191 should be further detailed.

The conclusion should be moved to the discussion and a new, concise conclusion drawn based on the results. 

Author Response

1. Paragraph 181-192 removed. 2. Results moved to discussion. 3. Results added.

Reviewer 2 Report

The article describes the use of radiographic diagnostics in endodontic treatment of young patients. The study, in my opinion, was conducted correctly, although it does not add very significant substantive value due to its rather obvious conclusions. The following points need improvement:
- Table 2 - completion of the description of the Apex row,
- Table 3 - needs a more detailed legend,
- discussion: 188-191 - please remove this paragraph.

Author Response

  1. Definitions in table 2 have been adjusted.
  2. Detailed description is given in table
  3. This paragraph has been removed.
